# Laparoscopic Ventriculoperitoneal Shunt Insertion Without a Peel-Away Sheath in Children: A Comparison with Conventional Open Surgery

**DOI:** 10.3390/children13010072

**Published:** 2026-01-02

**Authors:** Miri Ryu, Ayoung Kang, Soo-Hong Kim, Jae Hun Chung, Hanpyo Hong, Soon-Ki Sung

**Affiliations:** 1Department of Surgery, Pusan National University Yangsan Hospital, Yangsan 50612, Republic of Korea; bityeoul@daum.net (M.R.); 89april3@hanmail.net (A.K.); drjh1013@gmail.com (J.H.C.); 2School of Medicine, Pusan National University, Yangsan 50612, Republic of Korea; nscastle@naver.com; 3Division of Pediatric Surgery, Pusan National University Children’s Hospital, Yangsan 50612, Republic of Korea; 4Research Institute for Convergence of Biomedical Science and Technology, Pusan National University Yangsan Hospital, Yangsan 50612, Republic of Korea; honghanpyo@pnuyh.co.kr; 5Department of Neurosurgery, Pusan National University Yangsan Hospital, Yangsan 50612, Republic of Korea

**Keywords:** ventriculoperitoneal shunt, laparoscopy, pediatric, hydrocephalus

## Abstract

**Highlights:**

**What are the main findings?**
Laparoscopy-assisted VPS insertion without a peel-away sheath in children shows outcomes comparable to conventional open surgery.In cases with prior abdominal surgery, laparoscopy allows safe catheter placement by avoiding or lysing adhesions.

**What is the implication of the main finding?**
Laparoscopic VPS insertion without a peel-away sheath is a safe and effective surgical option in pediatric patients.Particularly in revisions or patients with previous operations, it may help reduce adhesion-related shunt failure, although further studies are warranted.

**Abstract:**

**Background/Objectives**: Hydrocephalus is primarily treated with open ventriculoperitoneal shunt (VPS) insertion, but laparoscopy-assisted VPS insertion has emerged as an alternative. This study compared outcomes and complications of laparoscopic versus open VPS insertion without a peel-away sheath in pediatric patients. **Methods**: A retrospective review was conducted on 121 VPS insertions (2012–2025) at a tertiary pediatric center in Korea. Patients were categorized into laparoscopic (*n* = 42) and open (*n* = 79) groups. The laparoscopic technique utilized only standard reusable instruments, without a peel-away sheath. Demographics, surgical parameters, postoperative recovery, and unplanned revision rates were analyzed. A Cox proportional hazards regression model was used to evaluate catheter survival, adjusting for baseline characteristics that differed significantly between the groups. Five-year shunt survival was assessed using Kaplan–Meier survival analysis. **Results**: The laparoscopic group had more revision cases and previous abdominal surgeries; additionally, intra-abdominal adhesions were more common (52.4% vs. 3.8%), and adhesiolysis was more often performed (9.5% vs. 1.3%). However, no differences were found in total operative time, distal catheter insertion time, or perioperative complications. After adjusting for demographic differences between the groups, Cox regression analysis demonstrated no significant difference in catheter survival. Both short-term (12 months) and long-term (5 years) shunt survival rates were comparable between the groups. **Conclusions**: Laparoscopic VPS insertion without a peel-away sheath is feasible and safe in pediatric patients, including those with complex surgical histories. It offers favorable recovery and adhesion management outcomes without compromising shunt durability, supporting its use as a practical alternative in resource-limited settings.

## 1. Introduction

Hydrocephalus is caused by various conditions and is primarily treated with a ventriculoperitoneal shunt (VPS) as the standard intervention [1]. Laparotomy has traditionally been the standard approach for VPS placement. However, since the introduction of the laparoscopic approach by Armbruster in 1993, minimally invasive techniques have emerged as viable alternatives [2].

Laparoscopy-assisted VPS insertion offers direct visualization of the peritoneal cavity, which may reduce the risk of surgical morbidity, including incisional hernias and postoperative adhesions commonly associated with open surgery [3]. Catheter failure, including malfunction, infection, misplacement, or distal dislocation, frequently necessitates revision surgeries. In pediatric patients, repeated distal catheter revisions are often required due to ongoing physical growth and long-term shunt dependency, complicating long-term management [4]. Moreover, reoperations can be technically challenging due to adhesions and limited access through prior laparotomy sites [5]. Therefore, the laparoscopic approach may provide significant advantages in pediatric VPS surgeries [6,7].

While laparoscopic VPS surgery has been increasingly adopted in pediatric neurosurgery worldwide, comparative evidence in children remains limited. A recent systematic review and meta-analysis reported comparable distal revision rates between laparoscopic and open peritoneal catheter placement in children [8]. However, most reported techniques rely on specialized disposable introducers such as peel-away sheaths [6,9,10], which may not be readily available or reimbursed in all healthcare systems. In this study, we utilized conventional laparoscopic instruments without a peel-away sheath, demonstrating a technique that may be more broadly applicable in resource-constrained settings.

This study evaluated the clinical outcomes and complication profiles of laparoscopic vs. open VPS procedures in Korean pediatric patients and assessed the feasibility and safety of a sheath-free laparoscopic approach.

## 2. Materials and Methods

### 2.1. Materials

From January 2012 to January 2025, 124 VPS insertions were performed in 80 patients under 17 years of age at Pusan National University Children’s Hospital (Busan, Republic of Korea). Patients requiring additional major abdominal procedures with prolonged operative time, such as Nissen fundoplication, incisional hernia repair, or umbilical hernia repair, were excluded. Three patients met these exclusion criteria, resulting in 121 VPS insertions in 77 patients included in the analysis (Figure 1).

### 2.2. Methods

A retrospective review of medical records was conducted, and patients were divided into two groups based on the surgical approach for peritoneal catheter placement. The laparotomy group included patients in whom the distal catheter was inserted into the peritoneal cavity through a conventional open approach, whereas in the laparoscopic group, the catheter was placed using a laparoscopic technique. Abdominal access was performed by a single surgeon in all cases.

The following parameters were analyzed: patient demographics, including gestational age, birth weight, age and weight at surgery, and sex; causes of hydrocephalus; history of prior abdominal surgery; total operative time; operative time for distal catheter insertion; presence of intra-abdominal adhesions and whether adhesiolysis was performed; length of hospital stay; time to diet resumption; shunt duration; complications; and unplanned revisions. In this study, unplanned revision was defined as any revision surgery performed within 1 year of the initial VPS insertion. For cases categorized as distal catheter malfunction, postoperative imaging was additionally reviewed to confirm distal catheter migration or abnormal tip position.

Statistical analyses were performed using SPSS (version 28; IBM Corp., Armonk, NY, USA). Continuous variables were compared using the independent t-test, and categorical variables were analyzed using the chi-square test. A Cox proportional hazards regression model was used to compare catheter survival between the two groups. The model was adjusted for baseline characteristics that differed between the groups, including age, height, weight, and a history of previous abdominal surgery. Hazard ratios (HRs) with 95% confidence intervals (CIs) were calculated from the model. Kaplan–Meier survival analysis was used to compare catheter survival in the two groups for up to 5 years. Because the follow-up durations differed between the laparotomy and laparoscopic groups, survival outcomes were standardized to a 5-year period for comparison. In addition, since catheter replacement may become necessary beyond 5 years due to patient growth, shunt survival was assessed up to 5 years. A *p*-value of <0.05 was considered statistically significant.

### 2.3. Operation Procedure

#### 2.3.1. Laparotomy

Under general anesthesia, the patient was placed in the supine position, and skin preparation and draping were performed by the neurosurgery team. A transverse skin incision was made along the natural skin crease in the left or right upper quadrant, followed by peritoneal opening. A purse-string suture was applied to the peritoneum, and the distal catheter (approximately 25–30 cm) was inserted into the peritoneal cavity. The incision was subsequently closed.

#### 2.3.2. Laparoscopy

Under general anesthesia, the patient was positioned supine, and skin preparation and draping were performed by the neurosurgery team. Two different techniques were used depending on the patient’s weight.

For patients weighing < 10 kg, one 3 mm optical port and two 3 mm working ports were used. Under direct visualization, a grasper inserted through the right (or left) lower quadrant port was passed through the right (or left) upper quadrant port. The distal catheter tip of the shunt was then pulled into the abdominal cavity and positioned appropriately (Figure 2).

For patients weighing over 10 kg, a 10 mm optical port and 5 mm working port were used. Both a 5 mm camera and 5 mm grasper were inserted through the 10 mm port. A grasper inserted through the 5 mm port at the umbilicus was maneuvered through the right (or left) upper quadrant port. The distal catheter was then advanced into the abdominal cavity under direct visualization and placed in the desired position (Figure 3).

The operative technique using reusable trocars is demonstrated in Appendix A.

## 3. Results

The overall patient demographics are summarized in Table 1. No significant differences were observed in gestational age, birth weight, or sex between the two groups. At surgery, patients in the laparoscopic group tended to be older and have greater body weight and height than did those in the laparotomy group.

Hemorrhage-induced hydrocephalus was the most common indication in both groups (46.8% vs. 66.7%), followed by congenital anomalies and infection-related hydrocephalus. No significant difference was determined in the overall distribution of indications between the groups (*p* = 0.154; Table 2).

The proportion of primary shunt operations was higher in the laparotomy group (58.2%), whereas the laparoscopic group included a greater proportion of patients undergoing second or subsequent VPS insertions (38.1% vs. 58.2%, *p* = 0.035). The number of patients who underwent three or more VPS procedures was also higher in the laparoscopic group. With respect to abdominal surgical history, no significant difference was observed between the groups when prior VPS procedures were excluded (*p* = 0.978). However, when previous VPS insertions were included, the laparoscopic group showed a significantly higher rate of prior abdominal surgeries, with 71.4% having undergone previous operations compared to 46.8% in the laparotomy group (*p* = 0.009; Table 3).

No significant differences were observed between the groups in total operative time (*p* = 0.443) or distal catheter insertion time (*p* = 0.572). Intra-abdominal adhesions were more frequently observed in the laparoscopic group, affecting 52.4% of patients, than in the laparotomy group at 3.8% (*p* < 0.001) (Figure 4). Adhesiolysis was performed in 9.5% of laparoscopic and 1.3% of laparotomy cases (*p* < 0.001). To avoid placing the distal catheter in areas with intra-abdominal adhesions, the insertion site was changed in 21.4% of patients in the laparoscopic group, whereas no changes were made in the laparotomy group (*p* < 0.001). Patients in the laparoscopic group had a significantly shorter hospital stays (*p* < 0.001) and resumed feeding earlier (*p* = 0.006). No periprocedural complications, such as bowel injury, were reported in either group (Table 4).

The rate of unplanned revisions within 12 months after surgery did not differ significantly between the two groups. Distal catheter malfunction was the most common cause of shunt failure, followed by proximal malfunction and infection. Although the total rate of shunt failure was higher in the laparotomy group, the difference was not significant (*p* = 0.132).

The causes of shunt failure during the entire follow-up period are presented in Table 5. In both groups, malfunction was the most frequent cause, with distal malfunction being the most common subtype. Among patients with distal catheter malfunction, catheter migration was observed in 7 patients (46.7%) in the laparotomy group and in 3 patients (75%) in the laparoscopy group, with no statistically significant difference in incidence between two groups (*p* = 0.394). Infections were observed more often in the laparoscopy group (30.0% vs. 9.7%, *p* = 0.119), but the difference was not significant. Notably, the *p*-value of 0.009 reflects the difference in the overall incidence of shunt failure between the two groups, rather than differences in the distribution of specific causes.

As baseline characteristics were differed between the two groups, Cox proportional hazards regression analysis was performed for adjustment. The analysis showed no significant association between surgical approach and shunt survival (HR, 1.00; 95% CI, 0.53–1.88; *p* = 0.999). Other variables, including age, weight, height, and previous abdominal operation history were also not significantly associated with catheter survival (Figure 5).

The median shunt survival time was 2035 d in the laparotomy group and 2195 d in the laparoscopy group. Kaplan–Meier analysis showed no significant difference in 5-year shunt survival between the two approaches (*p* = 0.638; Figure 6). The follow-up period was 1286.95 days in the laparotomy group and 717.45 days in the laparoscopic group (Table 4).

Among the 77 children, 15 died during the follow-up period. Of these, two deaths were directly related to shunt complications. One patient died due to repeated shunt infections following an initial open surgery; both the initial insertion and revision were performed via an open approach. The other died from hydrocephalus caused by a fracture of a shunt that had been initially inserted via an open approach; the subsequent procedures were performed laparoscopically. The remaining 13 deaths were attributed to underlying medical conditions or unrelated causes, including respiratory complications or other systemic conditions. One patient died of an unknown cause.

## 4. Discussion

Laparoscopic VPS insertion offers various potential advantages over traditional open surgery. One of the best-known benefits is its favorable cosmetic outcome, owing to smaller and fewer incisions [5,11,12]. In our experience, as children grow, previously used upper abdominal incisions tend to shift cephalad relative to changing body proportions. In some cases, this renders the original incision site unsuitable for reuse in subsequent procedures. Consequently, new incisions are often made slightly lower than the previous one, resulting in the formation of so-called “step-ladder” scars—multiple transverse scars aligned vertically, similar to the rungs of a ladder. This pattern not only raises cosmetic concerns but may also restrict future surgical access due to scarring or tissue distortion [13]. Notably, laparoscopic surgery minimizes both the size and number of incisions, improving cosmetic outcomes and potentially reducing postoperative pain compared to that of open surgery [13,14].

Recent systematic reviews and meta-analyses in children have suggested comparable distal revision rates between laparoscopic and open distal catheter placement [8,15]. Based on our results, laparoscopic VPS insertion is a technically advantageous option in revision surgeries or in cases complicated by intra-abdominal adhesions. Open abdominal surgery inevitably causes intra-abdominal adhesions [16], thereby complicating subsequent procedures. One of the reasons we adopted laparoscopy for revision cases was our previous experience with such complications. Laparoscopy offers greater flexibility in dealing with adhesions. In open surgery, encountering adhesions through a previous incision often necessitates a larger incision to ensure adequate adhesiolysis and catheter placement. In contrast, laparoscopy allows adhesiolysis to be performed through additional ports even in cases of severe adhesions and enables direct placement of the catheter in an adhesion-free area.

In the laparoscopic group, some patients required additional intra-abdominal maneuvers, such as adhesiolysis (9.5%) or inserting catheters away from adhesion areas (21.4%). Despite these technically demanding situations, catheter survival was comparable to that of the laparotomy group. Because such maneuvers are difficult to perform during laparotomy, laparotomy outcomes might have been less favorable. From this standpoint, laparoscopy may offer a meaningful advantage when managing complex intra-abdominal conditions.

Laparoscopic surgery is generally associated with faster recovery and shorter hospital stays [17]. In our study, we also observed that patients in the laparoscopic group had shorter hospital stays and earlier initiation of feeding compared to those in the laparotomy group. This may reflect one of the inherent advantages of the laparoscopic approach. However, this difference should be interpreted with caution. Laparoscopic surgery was introduced more recently at our institution, coinciding with broader institutional changes such as the adoption of enhanced recovery after surgery protocols [18,19]. These system-wide changes may have independently contributed to improved recovery metrics. Additionally, hospital stay and feeding resumption can be influenced by various patient-specific factors, including comorbidities, nutritional status, and social circumstances. Therefore, attributing these findings solely to the surgical approach itself is difficult.

Furthermore, laparoscopic surgery is generally associated with a lower risk of postoperative adhesion formation compared to that of open procedures [20,21]. This characteristic may have important implications for long-term management. When laparoscopy is used for the initial VPS placement, future revisions may be performed in a less adhesive environment, potentially reducing surgical difficulty and complication risk. Although our study did not directly evaluate adhesion severity during reoperation, this hypothesis warrants further investigation in larger, prospective studies.

Although the peel-away sheath is commonly used in laparoscopic VPS procedures [9,10], it was not employed in this study. In our setting, this may appear as a deviation from standard practice, but it reflects an adaptation to local healthcare constraints. In Korea, the peel-away sheath is not reimbursed by the national health insurance system, limiting its routine use. While this reflects a constraint specific to our healthcare system, it also underscores the importance of adaptable techniques suitable for resource-limited or reimbursement-restricted environments. As patients must pay for this device out of pocket, its use may lead to hesitation or discomfort during the decision-making process. To address this, we adopted a technique using a reusable trocar that was already employed in other laparoscopic procedures. In our experience, this approach reduced financial burden and provided several practical surgical advantages. It allowed partial adhesiolysis, enabled direct visualization, and facilitated flexible catheter positioning through additional ports. These advantages may support the use of reusable trocars as a viable alternative in settings with limited access to single-use devices. Because we did not use a peel-away sheath, we used a grasper to manipulate and position the distal catheter tip with greater flexibility. This allowed us to place the catheter in an adhesion-free area under direct visualization, which we believe contributed to the favorable postoperative outcomes.

Initially, only open surgery was performed at our institution. Following difficulties with catheter insertion and complications such as pseudocysts, we began performing laparoscopic VPS insertion for revision cases whenever feasible. With growing surgical experience, we gradually extended its application to primary VPS procedures. Consequently, patients in the laparoscopic group were older and had greater height and weight than those in the laparotomy group. These characteristics generally reflect favorable surgical conditions for laparoscopy. However, the laparoscopic group also included a higher proportion of revision cases, which are typically more complex due to adhesions or prior complications. This duality—favorable physical attributes but greater clinical complexity—should be carefully considered when interpreting the results.

In contrast, the laparotomy group had a longer follow-up period and catheter survival duration, simply because open surgery had been introduced earlier. This may have created the impression that open surgery yielded more durable outcomes. To account for differences in group composition and follow-up time, we compared the 5-year catheter survival rate rather than the total duration, which revealed no significant difference between the two groups. This approach allowed for a more balanced and equitable comparison across groups with inherently different clinical backgrounds and follow-up periods.

Although the laparoscopic group included more revision cases—patients who are more likely to present with adhesions and increased surgical complexity—no significant difference was determined in short-term complications or 5-year survival rate compared to that of the laparotomy group. This suggests that the laparoscopic approach may be useful even in more challenging surgical contexts.

This study has some limitations. First, there is an inherent time bias due to the chronological introduction of the surgical techniques. Open surgery was introduced earlier, resulting in a longer follow-up duration and potentially more opportunities for complications or revisions to be observed. Although we attempted to minimize this effect by comparing 5-year catheter survival, residual bias cannot be completely excluded. In addition, laparoscopic procedures were introduced later in the study period; therefore, a learning-curve effect may have influenced operative metrics and outcomes. Additionally, this was a retrospective, single-center analysis with a relatively small sample size, and as such, statistical power was limited. Larger, multicenter, prospective studies are warranted to further validate these findings.

## 5. Conclusions

In conclusion, laparoscopic VPS insertion in pediatric patients demonstrated surgical time and complication rates comparable to those of conventional laparotomy. Despite a higher proportion of patients undergoing second or subsequent operations—who are typically at increased risk of complications—in the laparoscopy group, no significant difference was observed in catheter survival between the two groups over the same follow-up period. These findings suggest that laparoscopic VPS insertion without a peel-away sheath is a safe and effective alternative to open surgery for pediatric patients.

## Figures and Tables

**Figure 1 children-13-00072-f001:**
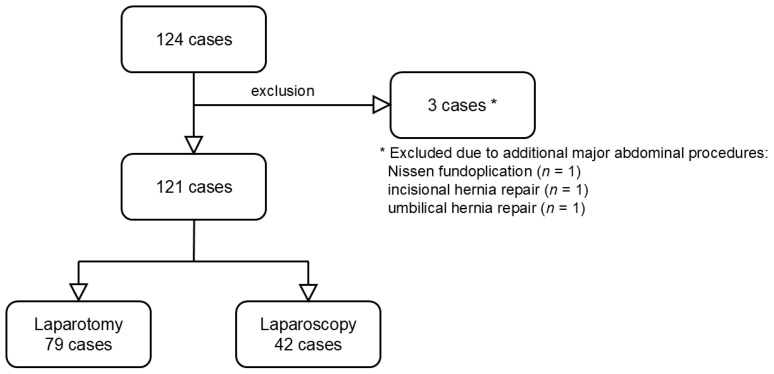
Flow diagram. Overall, 124 cases were screened. Three were excluded due to concurrent abdominal surgery. Among the remaining 121 cases, 79 and 42 underwent open and laparoscopic VPS insertion, respectively.

**Figure 2 children-13-00072-f002:**
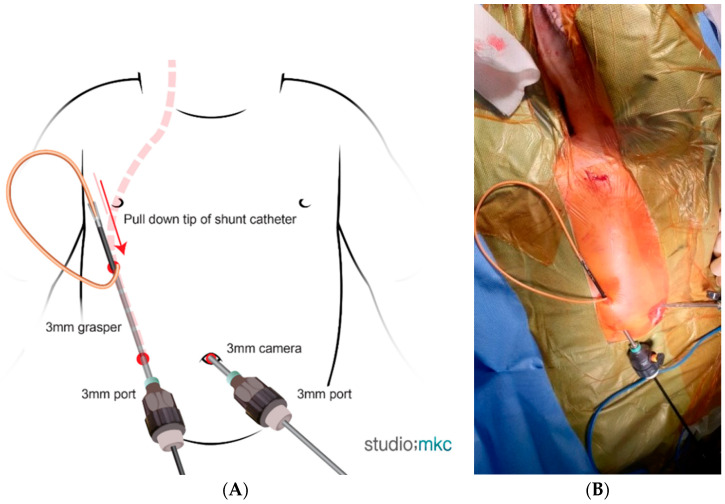
Laparoscopic VPS insertion in patients weighing < 10 kg. A 3 mm optical port was placed at the umbilicus, and two 3 mm working ports were inserted in the upper and lower quadrants. Under direct visualization, a grasper inserted through the lower quadrant port was passed through the upper quadrant port. The distal tip of the shunt catheter was then pulled into the abdominal cavity and positioned appropriately. (**A**) Schematic illustration. (**B**) Intraoperative view.

**Figure 3 children-13-00072-f003:**
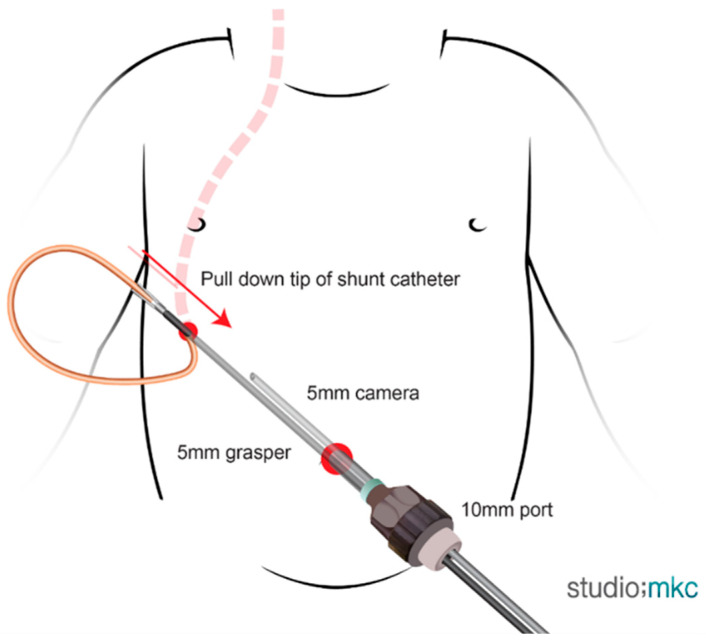
Laparoscopic VPS insertion in patients weighing ≥ 10 kg. A 10-mm optical port was placed at the umbilicus and 5-mm working port was inserted in the upper quadrant. Both a 5-mm camera and 5-mm grasper were inserted through the 10-mm port. The grasper, introduced through the 5-mm port, was maneuvered to grasp the distal tip of the shunt catheter. Under direct visualization, the catheter was pulled into the abdominal cavity and positioned appropriately.

**Figure 4 children-13-00072-f004:**
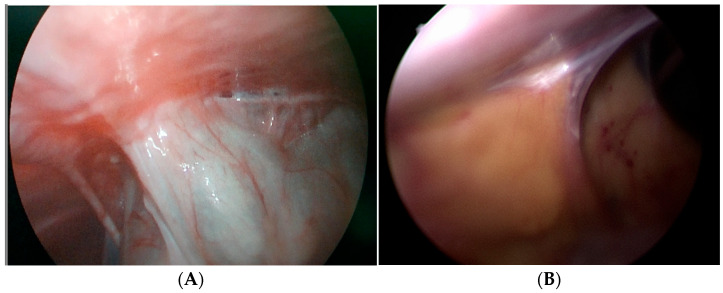
Laparoscopic findings of intra-abdominal adhesions. (**A**) Dense adhesions were noted at the previous midline insertion site; therefore, the catheter was placed in the right upper quadrant. The patient had a history of three prior VPS surgeries. (**B**) Bowel and omental adhesions were observed at the previous insertion site, making it unsuitable for reuse. A new insertion site was selected at a more lateral location to avoid the adhesion area.

**Figure 5 children-13-00072-f005:**
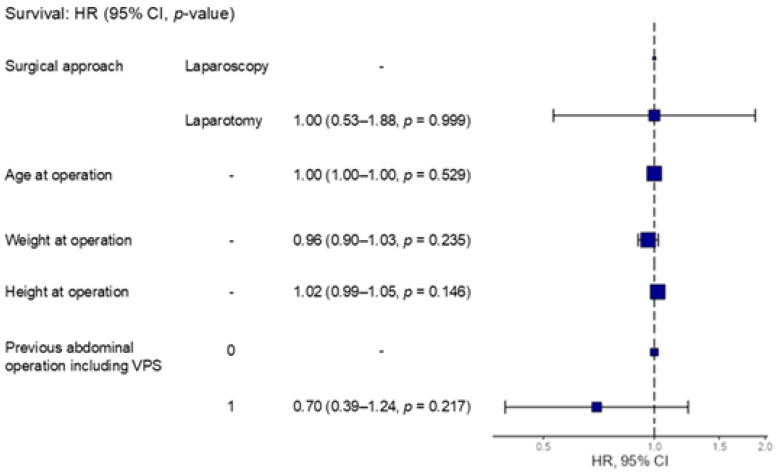
Forest plot of Cox proportional hazards regression analysis of shunt survival. In the multivariable Cox regression analysis, the surgical approach (laparotomy versus laparoscopy) was not significantly associated with shunt survival (HR, 1.0; 95% CI, 0.53–1.88; *p* = 0.999). Similarly, age, weight, and height at the time of surgery did not have a significant effect on the outcome. History or previous abdominal surgery was also not significantly associated with a decreased risk of shunt failure (HR, 0.70; 95% CI, 0.39–1.24; *p* = 0.217).

**Figure 6 children-13-00072-f006:**
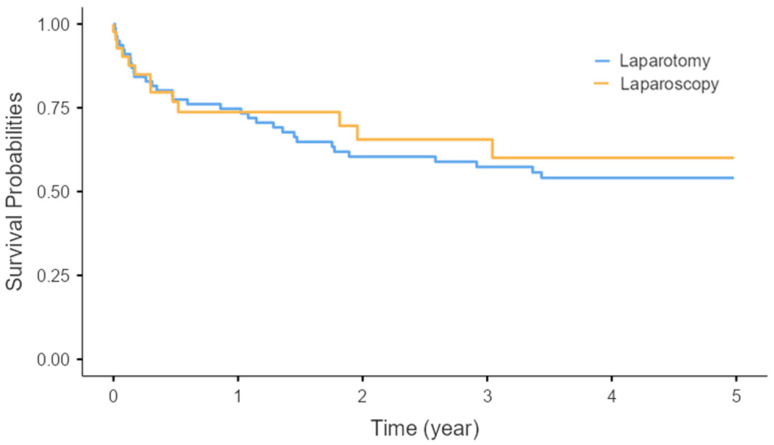
Kaplan–Meier analysis of 5-year shunt survival. No significant differences were observed in 5-year shunt survival between the laparotomy and laparoscopy groups (*p* = 0.638).

**Table 1 children-13-00072-t001:** Patient demographics.

	Laparotomy (*n* = 79)	Laparoscopy (*n* = 42)	*p*-Value
Gestational age (week)	31.73 ± 7.51	32.67 ± 5.52	0.487
Gender (M/F)	43 (54.6%)/36 (45.6%)	24 (57%)/18 (43%)	
Birth weight (kg)	2.05 ± 1.04	2.07 ± 1.08	0.904
Age (day) at operation	1036.52 ± 1414.89 (7 days–16 years)	1795.98 ± 1514.87 (14 days–13 years)	0.007
Weight (kg) at operation	11.83 ± 11.66 (1.8–58.9)	17.06 ± 12.09 (2.6–50.7)	0.022
Height (cm) at operation	77.63 ± 30.64 (43–173)	95.01 ± 32.08 (44–155)	0.004

Data are presented as means ± SD (min–max) or *n* (%).

**Table 2 children-13-00072-t002:** Causes of hydrocephalus (indications for VPS insertion).

	Laparotomy (*n* = 79)	Laparoscopy (*n* = 42)	Total (*n* = 121)
Hemorrhage induced	37 (46.8%)	28 (66.7%)	65 (53.7%)
Tumorous condition induced	7 (8.9%)	1 (2.4%)	8 (6.6%)
Congenital anomaly related	16 (20.3%)	6 (14.3%)	22 (18.2%)
Infectious condition related	7 (8.9%)	4 (9.5%)	11 (9.1%)
Hypoxia-induced encephalopathy related	7 (8.9%)	0	7 (5.8%)
Unknown origin	5 (6.3%)	3 (7.1%)	8 (6.6%)

Data are presented as *n* (%). *p*-value = 0.154.

**Table 3 children-13-00072-t003:** Operation history.

	Laparotomy (*n* = 79)	Laparoscopy (*n* = 42)	*p*-Value
No. of VPS insertion			0.035
Primary	46 (58.2%)	16 (38.1%)	
2nd	27 (34.2%)	12 (28.6%)	
3rd	5 (6.3%)	9 (21.4%)	
More than 3rd time	1 (1.3%)	5 (11.9%)	
Previous abdominal surgery except previous VPS insertion			0.978
Yes	12 (15.2%)	7 (16.7%)	
No	67 (84.8%)	35 (83.3%)	
Previous abdominal surgery include previous VPS insertion			0.009
Yes	37 (46.8%)	30 (71.4%)	
No	42 (53.2%)	12 (28.6%)	

Data are presented as *n* (%). VPS, ventriculoperitoneal shunt.

**Table 4 children-13-00072-t004:** Operative findings and outcomes.

	Laparotomy (*n* = 79)	Laparoscopy (*n* = 42)	*p*-Value
Overall OP time (min)	73.61 ± 26.98 (30–175)	69.16 ± 35.63 (15–200)	0.443
Distal catheter insertion time (min)	34.06 ± 20.48 (10–120)	31.79 ± 22.14 (5–140)	0.572
Overall OP time excluding adhesiolysis cases (min) ^†^	73.78 ± 27.11 (30–175)	70.53 ± 36.28 (30–200)	0.589
Distal catheter insertion time excluding adhesiolysis cases (min) ^†^	33.92 ± 20.57 (10–120)	30.53 ± 22.08 (5–140)	0.417
Intraoperative complication	0	0	
Recognition of adhesion	3 (3.8%)	22 (52.4%)	<0.001
Adhesiolysis	1 (1.3%)	4 (9.5%)	<0.001
Inserting away from adhesion	0	9 (21.4%)	<0.001
Hospitalization (day)	19.73 ± 28.3 (2–166)	8.05 ± 7.57 (0–36)	<0.001
Diet start (day)	0.53 ± 0.73 (0–3)	0.24 ± 0.43 (0–1)	0.006

Data are presented as means ± SD (min–max). ^†^ *n* = 121 (laparotomy: 79 cases, laparoscopy: 42 cases).

**Table 5 children-13-00072-t005:** Causes of shunt failure.

	Laparotomy (*n* = 79)	Laparoscopy (*n* = 42)	*p*-Value
Malfunction			
Proximal	9 (29.0%)	2 (20%)	0.581
Distal	15 (48.4%)	4 (40%)	0.647
Unknown	3 (9.7%)	1 (10%)	0.978
Infection ^†^	3 (9.7%)	3 (30%)	0.119
Pseudocyst	1 (3.2%)	0 (0.0%)	0.572
Total failure	31 (100%)	10 (100%)	0.090
VPS related death ^†^	1 (1.3%)	1 (2.4%)	0.656

Data is presented as the *n* (%). ^†^ VPS-related deaths are included as infections.

## Data Availability

The data presented in this study are not publicly available due to institutional and national regulations, as well as the conditions of the Institutional Review Board approval, on sharing patient-level clinical data. De-identified data that support the findings of this study are available from the corresponding author on reasonable request.

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
