# Peer review of "Laparoscopic Ventriculoperitoneal Shunt Insertion Without a Peel-Away Sheath in Children: A Comparison with Conventional Open Surgery"

_children, 2026, doi:10.3390/children13010072_

Round 1
Reviewer 1 Report
Comments and Suggestions for Authors
This study compares laparoscopic ventriculoperitoneal shunt insertion without a peel-away sheath in children with conventional open surgery. The authors report non-inferior outcomes and suggest potential advantages of the laparoscopic approach despite a higher proportion of repeat insertions in the laparoscopic group. Additionally, I consider the ability to detect adhesions and modify the catheter insertion site an important potential benefit and a key outcome.
However, the manuscript should clarify its main message: is the primary contribution the “sheath-free” technique itself, or the general advantages of laparoscopic shunt placement? Ideally, a comparison between laparoscopic techniques with vs without a peel-away sheath would best address the stated novelty; although I understand this may be difficult in a single-center study.
Because recent systematic reviews/meta-analyses have already summarized the benefits and limitations of laparoscopic versus open VP shunt placement, these should be cited in the Introduction and/or Discussion. The Discussion should then focus more specifically on what is new about the sheath-free approach. If the authors intend the main argument to be “laparoscopy vs open,” they should explicitly state what incremental novelty this study adds beyond existing reviews.
Author Response
|
Comments 1: This study compares laparoscopic ventriculoperitoneal shunt insertion without a peel-away sheath in children with conventional open surgery. … However, the manuscript should clarify its main message: is the primary contribution the “sheath-free” technique itself, or the general advantages of laparoscopic shunt placement? |
|
Response 1: Thank you for this important comment. To clarify that the primary contribution of our study is the peel-away sheath-free laparoscopic distal catheter insertion technique, rather than the general advantages of laparoscopy, we revised the wording in the Highlights (lines 16 and 21) and added/strengthened relevant statements in the Discussion (lines 374–378) to align the main message accordingly. Given that our comparator was open surgery, we briefly acknowledge existing evidence in the Discussion and emphasize the incremental novelty of our work, namely the sheath-free approach.
|
|
Comments 2: Because recent systematic reviews/meta-analyses have already summarized the benefits and limitations of laparoscopic versus open VP shunt placement, these should be cited in the Introduction and/or Discussion. The Discussion should then focus more specifically on what is new about the sheath-free approach. If the authors intend the main argument to be “laparoscopy vs open,” they should explicitly state what incremental novelty this study adds beyond existing reviews. Response 2: As suggested, we added recent systematic review/meta-analysis references in both the Introduction and the Discussion (new References #8 and #15; Introduction lines 66–68 and Discussion lines 324–325). Building on this context, we revised the Discussion to more clearly describe the added contribution of our study, focusing on the practical advantages of the sheath-free technique. |
Reviewer 2 Report
Comments and Suggestions for Authors
Dear Author
Following are my comments
- The surgeon can be confounding and the study is retrospective, was the surgeon the same for both laparoscopic versus open VPS insertion?
- The reason for exclusion of 3 cases is needed
- The manuscript needs the ethical committee confirmation and signing the written informed consent by parents of patients
- Some main complications should be consider infection, abdominal pain, catheter migration, and rare intraoperative injuries.
Author Response
|
Comments 1: The surgeon can be confounding and the study is retrospective, was the surgeon the same for both laparoscopic versus open VPS insertion? |
|
Response 1: Yes. In our cohort, abdominal access and distal catheter placement for all cases in both groups were performed by a single surgeon, thereby minimizing inter-surgeon variability. We clarified this in the Methods section (lines 94–95). In addition, we strengthened the Limitations section to acknowledge potential time-related confounding, including a learning-curve effect, because laparoscopic procedures were introduced later in the study period (lines 405–407).
|
|
Comments 2: The reason for exclusion of 3 cases is needed. |
|
Response 2: The exclusion criteria (concurrent major abdominal procedures leading to prolonged operative time) were described in the Materials/Methods (lines 80–83). In response to the reviewer’s comment, we additionally revised Figure 1 to explicitly state the reasons for excluding the three cases, so that the flow diagram is self-explanatory.
Comments 3: The manuscript needs the ethical committee confirmation and signing the written informed consent by parents of patients. Response 3: This study was approved by the institutional review board (IRB statement, lines 429–431). Because this was a retrospective chart review, written informed consent was waived, as stated in the manuscript (informed consent statement, lines 432–433), in accordance with the MDPI-required format.
Comments 4: Some main complications should be consider infection, abdominal pain, catheter migration, and rare intraoperative injuries. Response 4: No major intraoperative complications were observed (as described in the manuscript and Table 4). For catheter migration among cases categorized as distal catheter malfunction, we additionally reviewed postoperative imaging and reported the migration-related findings in the Results (lines 259–262). Postoperative abdominal pain was not included as a comparative endpoint because reliable and consistent assessment was difficult in a substantial proportion of our cohort, many of whom were neurologically impaired children.
|
Round 2
Reviewer 1 Report
Comments and Suggestions for Authors
The authors have appropriately revised the manuscript in response to the reviewers’ comments, resulting in an improvement in the content.